# Involvement of Mitochondrial Mechanisms and Cyclooxygenase-2 Activation in the Effect of Desethylamiodarone on 4T1 Triple-Negative Breast Cancer Line

**DOI:** 10.3390/ijms23031544

**Published:** 2022-01-28

**Authors:** Ferenc Gallyas, Fadi H. J. Ramadan, Kitti Andreidesz, Eniko Hocsak, Aliz Szabo, Antal Tapodi, Gyongyi N. Kiss, Katalin Fekete, Rita Bognar, Arpad Szanto, Zita Bognar

**Affiliations:** 1Department of Biochemistry and Medical Chemistry, University of Pecs Medical School, 7624 Pecs, Hungary; ferenc.gallyas@aok.pte.hu (F.G.J.); fadi.ramadan@aok.pte.hu (F.H.J.R.); andreidesz.kitti@pte.hu (K.A.); Eniko.hocsak@aok.pte.hu (E.H.); aliz.szabo@aok.pte.hu (A.S.); antal.tapodi@aok.pte.hu (A.T.); gyongyi.nagyne@aok.pte.hu (G.N.K.); Katalin.fekete@aok.pte.hu (K.F.); rita.bognar@aok.pte.hu (R.B.); 2Szentagothai Research Centre, University of Pecs, 7624 Pecs, Hungary; 3LERN-UP Nuclear-Mitochondrial Interactions Research Group, 1245 Budapest, Hungary; 4Urology Clinic, UP Medical Center, University of Pecs Medical School, 7624 Pecs, Hungary; szanto.arpad@pte.hu

**Keywords:** amiodarone, apoptosis, colony formation, invasive growth, Akt pathway, ΔΨm, therapy resistance, mitochondrial fragmentation, Seahorse

## Abstract

Novel compounds significantly interfering with the mitochondrial energy production may have therapeutic value in triple-negative breast cancer (TNBC). This criterion is clearly fulfilled by desethylamiodarone (DEA), which is a major metabolite of amiodarone, a widely used antiarrhythmic drug, since the DEA previously demonstrated anti-neoplastic, anti-metastasizing, and direct mitochondrial effects in B16F10 melanoma cells. Additionally, the more than fifty years of clinical experience with amiodarone should answer most of the safety concerns about DEA. Accordingly, in the present study, we investigated DEA’s potential in TNBC by using a TN and a hormone receptor positive (HR+) BC cell line. DEA reduced the viability, colony formation, and invasive growth of the 4T1 cell line and led to a higher extent of the MCF-7 cell line. It lowered mitochondrial transmembrane potential and induced mitochondrial fragmentation. On the other hand, DEA failed to significantly affect various parameters of the cellular energy metabolism as determined by a Seahorse live cell respirometer. Cyclooxygenase 2 (COX-2), which was upregulated by DEA in the TNBC cell line only, accounted for most of 4T1’s DEA resistance, which was counteracted by the selective COX-2 inhibitor celecoxib. All these data indicate that DEA may have potentiality in the therapy of TNBC.

## 1. Introduction

Breast cancer (BC) is the most frequent cancer type in women and the second primary cause of cancer-related death worldwide. Triple-negative (TN) form of the disease represents about 10–20% of all BC cases. Although of heterogeneous phenotype, TNBC is characterized by the lack of estrogen receptor (ER), progesterone receptor (PR), and human epidermal growth factor receptor (HER)2/neu gene amplification. TNBC patients typically present at a younger age a larger average tumor size, higher grade, and higher rates of lymph node positivity compared to patients with ER/PR-positive tumors [1]. Due to these features, TNBCs do not respond or became resistant to targeted therapies and, accordingly, have poor prognosis [2]. Furthermore, a progressive drug resistance leading to the formation of non-responding metastases often limits the systemic therapy [3,4]. Taxane- and anthracycline-based treatment represent the mainstream of TNBC chemotherapy, although optimization of the protocols is yet to be accomplished [4]. Previous studies suggest that there is a correlation between inflammation and tumor formation [5], and intensive research is trying to uncover the complex interaction between breast cancer and the immune system [6,7].

Cyclooxygenase-2 (COX-2) is an inducible form of the enzyme that catalyzes the synthesis of prostanoids, including prostaglandin E2 (PGE2), a major mediator of inflammation and angiogenesis. COX-2 is overexpressed in cancer cells and is associated with progressive tumor growth and resistance of cancer cells to conventional chemotherapy and radiation [8]. Furthermore, elevated expression of the COX-2 gene and activation of the COX-2/matrix metalloproteinase 1 pathway was implicated in brain metastasis formation of BC [9,10]. According to the current view, conventional chemotherapy and radiotherapy selects resistant cancer cells that are able to reinitiate tumor growth. There is compelling evidence of an active proliferative response, driven by increased COX-2 expression and PGE2 release, which contributes to the repopulation of the tumor and the resulting poor outcome for the patients [8]. Recent in vitro and in vivo pre-clinical studies have demonstrated that COX-2 overexpression plays a key role in tumor resistance by stimulating epithelial cell proliferation and angiogenesis, increasing multidrug resistance, and enhancing cell motility and invasion [8,11,12,13,14]. In TNBC, elevated COX-2 expression promotes stemness, indicating COX-2 as a potential therapeutic target [15]. However, clinical studies utilizing selective COX-2 inhibitors such as celecoxib in BC patients provided conflicting results indicating that molecular mechanisms associated with tumor development influenced the resistance to COX-2 inhibitors [16].

Based on their central role in metabolism and cell death regulation, mitochondria have emerged as novel anti-cancer therapeutic targets [17]. Mitochondria constitute a central hub in intermediary metabolism, are responsible for the majority of ATP synthesis, and regulate cell death via signaling molecules and the release of pro-apoptotic intermembrane proteins [18,19]. Cancer cells tend to use glycolysis for energy production even in the presence of sufficient oxygen supply to preserve intermediates for biosynthetic purpose [20]. However, cancers of the most aggressive phenotype such as drug-resistant and stem cancer cells rely on oxidative phosphorylation for energy production and reprogram their metabolism to meet the challenges of fast proliferation [17,20]. Accordingly, drugs that interfere with the mitochondrial functions disrupt this delicately balanced reprogrammed metabolism to an extent that is incompatible with survival and thus have strong potential in anti-cancer therapy [17,21].

Recent clinical trials combine immunotherapy with the traditional chemotherapy to combat therapy resistance [22]; however, effective management of TNBC patients remains a challenge in the clinical practice. On the other hand, the introduction of new drugs into the clinical practice is much delayed by the very lengthy safety protocols, which can be significantly shortened by repositioning a drug already approved for human therapy [23]. Desethylamiodarone (DEA) is the major metabolite of amiodarone (AM), an antiarrhythmic agent widely used in various life-threatening ventricular tachyarrhythmias. During antiarrhythmic therapy, AM accumulates rapidly in the lipophilic tissues, together with DEA [24], which is generated from the mother compound by the liver. Therapeutic AM plasma concentration is limited to 5.7 μM [25], since therapy-restricting side effects [26] develop in most cases at higher plasma concentrations. However, the accumulation of AM and DEA in tissues can result in concentrations exceeding the plasma concentration 100 to 1000 times [25]. Unlike AM, DEA is quite stable in the body, and it is eliminated with a half-life of about 40 days [27]. Except for the adipose tissue, its tissue accumulation exceeds that of AM [24,28,29]. However, at concentrations observable at therapeutic AM doses [24], DEA does not have therapy restricting side effects [30], which may hinder its introduction into clinical studies.

DEA’s accumulative and toxic properties suggested that it might have an anti-neoplastic potentiality. Furthermore, there is a vast clinical experience with the safety features of DEA gathered during the more than 50 years of AM’s therapeutic use [26]. That is, if DEA exerted a sufficiently effective anti-neoplastic effect at concentrations not exceeding those existing during AM therapy, it could be a readily available tool in cancer therapy. Previously, we reported DEA’s cytotoxicity in the bladder, cervix, and melanoma cell lines and its metastasis-limiting properties in a rodent lung metastasis model [31,32,33]. Furthermore, by using the B16F10 melanoma cell line, we demonstrated that DEA affected mitochondrial processes in live cells [34] similarly as it did previously in isolated liver and heart mitochondria [35]. A uniform anti-neoplastic effect on three very different, highly proliferative cell lines indicated that DEA interferes with fundamental survival-promoting processes likely present in most cancer cell lines of different tissue and species of origin. In the present report, we studied the anti-neoplastic and mitochondrial effects of DEA and combined the cytotoxic effect of DEA and the selective COX-2 inhibitor celecoxib on a TN line in comparison with a hormone receptor positive (HR+) BC line to identify potential differences in DEA’s molecular mechanisms in these two cancer types.

## 2. Results

### 2.1. DEA Induced Apoptotic Cell Death in BC Cell Lines

Previously, DEA demonstrated signs of anti-neoplastic potentiality in various cancer cell lines [31,32,33], including therapy-resistant lines such as T24 [32] and B16F10 [33]. Accordingly, we assessed its effect on the cell death process of the TN 4T1 BC line in comparison with the HR+ MCF-7 line of the less aggressive cancerous phenotype [36]. We used flow cytometry with the Muse™ Annexin V & Dead Cell Assay kit to investigate the mode of DEA-induced cell death. The assay utilizes cell surface annexin V binding that measures the appearance of phosphatidylserine on the external surface of the plasma membrane, which is a marker of apoptosis. A dead cell marker, 7-aminoactinomycin D (7-AAD), was also used as an indicator of cell membrane structural integrity. Late apoptosis is demonstrated by double positivity. We treated the cells with 5 and 10 µM DEA for 24 h before the flow cytometry analysis. In a concentration-dependent way, DEA induced early, then late apoptosis (Figure 1). At the lower DEA concentration, the rate of early apoptosis exceeded that of the late; however, at the higher drug concentration, late apoptosis predominated in both BC cell lines (Figure 1). However, we found a total apoptosis rate of 28.18 ± 6.34% for 5 and 56.05 ± 5.12% for 10 μM DEA in the 4T1 cell line in contrast to the apoptosis rates of 71.14 ± 6.39% for 5 and 78.94 ± 4.77% for 10 μM DEA observed in the MCF-7 cell line (Figure 1), indicating that MCF-7 cells were more sensitive toward DEA treatment than the 4T1 cells.

### 2.2. DEA Mitigated Invasive Growth of BC Cell Lines

Cell migration, an important aspect of cancer invasiveness, is often assessed by means of the wound-healing assay [37]. Accordingly, we inflicted a wound into semi-confluent monolayers of MCF-7 and 4T1 cells and treated them by 0, 5, or 10 µM DEA for 12 h. In accordance with its more malignant phenotype [38], cell migration of the TNBC cells was more intense, resulting in almost complete closing of the wound within 12 h, while the HR+ BC cell line achieved an about 40% healing during the same time. Ten µM DEA treatment completely prevented wound healing in the 4T1cell line, while it caused wound exacerbation due to killing cells at the wound edge in the MCF-7 cell line (Figure 2). At the concentration of 5 µM, DEA treatment induced a less pronounced effect on wound healing as it did at 10 µM (Figure 2).

To supplement the wound-healing assay, we assessed the invasive growth of the MCF-7 and 4T1 cells using an xCelligence Real-Time Cell Analysis (RTCA) system. The cells were cultured in the presence of 0, 5, or 10 µM DEA for 24 h and the cell index proportional to invasive growth of the cells was monitored in real time. DEA decreased invasive growth in a concentration-dependent manner in both MCF-7 and 4T1 cells (Figure 3). Similarly to the results of the wound-healing experiment, MCF-7 was more sensitive to the drug than 4T1 (Figure 3).

### 2.3. DEA Differentially Modulated Regulators and Markers of the Cell Death Process in the BC Cell Lines

To gain greater insight into the apoptosis pathway induced by DEA, we analyzed Akt activation, protein levels of Bcl-2 family members including pro-apoptotic Bad and Bax and antiapoptotic Bcl-2, and the caspase-3 cleavage and caspase-3-mediated cleavage of PARP, which are all reporters of the mitochondrial apoptotic pathway. As shown in Figure 4, the Akt phosphorylation at Ser^473^ decreased in a concentration-dependent manner, while the total Akt remained constant. This was accompanied by a significant decrease in the phosphorylation level of Bad at Ser^136^, while the overall level of Bad protein remained constant in both cell lines. We also found that DEA indeed resulted in a dose-dependent increase in the amount of p53 protein in 4T1 cells (Figure 4). We also measured a significant increase in Bax levels while Bcl-2 levels decreased (Figure 4). Furthermore, we found that DEA treatment led to an increase in the amount of a 19 kDa caspase-3 cleavage intermediate as well as cleaved poly (ADP-ribose) polymerase (PARP) (Figure 4). Taken together, these data demonstrate that cytotoxic effects of DEA on MCF-7 and 4T1 cells are due to the activation, at least partially, of two apoptotic pathways, the PI3K/Akt pathway and the mitochondrial pathway.

A differential expression of the therapy-resistance- and metastasis-promoting COX-2 gene among BC cell lines of various phenotypes has been reported previously [39]. Accordingly, we studied DEA’s effect on COX-2 protein levels in the 4T1 and MCF-7cell lines. In agreement with previous data [39], we detected considerable steady-state COX-2 levels in the TN BC cell line, while it was just above the detection limit in the HR+ one (Figure 3). Additionally, DEA increased the COX-2 level in a concentration-dependent way in the TN 4T1 cell line only (Figure 4).

### 2.4. DEA Caused the Loss of Mitochondrial Membrane Potential (ΔΨm)

Intact ΔΨm is pivotal for cellular survival due to its essential role in ATP synthesis, in providing the driving force for the transport of ions and proteins, and in mitochondrial quality control [40]. Therefore, we determined the effect of DEA on the ΔΨm of BC cells by using the positively charged fluorescent mitochondrial dye, JC-1. When the ΔΨm is normal, the dye accumulates in the mitochondria and forms J-aggregates that emit red fluorescence upon excitation. The aggregates disassemble, leaving green fluorescent monomers as the ΔΨm decreases, and the fluorescence disappears upon complete mitochondrial depolarization. We treated the MCF-7 and 4T1 BC cell lines for 3 h with 0, 5, or 10 µM DEA, before loading them with JC-1 and taking fluorescent microscopy images. At the concentration of 10 µM, the drug significantly depolarized the mitochondria in both BC cell lines, while 5 µM DEA did not have a considerable effect on the ΔΨm during the 3 h treatment (Figure 5).

### 2.5. DEA Induced Mitochondrial Fragmentation in BC Cell Lines

Healthy ΔΨm is required for the mitochondrial fusion; therefore, compromised ΔΨm often results in mitochondrial fragmentation. We studied the effect of DEA on mitochondrial network dynamics by fluorescent microscopy after loading the cells with MitoTracker Red to visualize the mitochondria. The MCF-7 and 4T1 cells were treated with 0, 5, or 10 µM of DEA for 3 h before the assessment of mitochondrial fragmentation. Similarly, as it did in melanoma cells previously [34], DEA treatment caused mitochondrial fragmentation in both BC cell lines in a concentration-dependent manner (Figure 6A).

Recently, a link has been established between the proliferation of cancer cells and mitochondrial fragmentation [42]. Accordingly, we performed immunoblot analysis of the proteins involved in the regulation of mitochondrial fusion and fission [43] from homogenates of BC cells treated identically to the fragmentation experiment in separate plates. In both BC cell lines, the DEA treatment increased the steady-state level of fusion-associated protein optic atrophy 1 (OPA1), but it decreased mitofusin (Mfn) 1 and 2 in 4T1 cells (Figure 6B). However, it increased the steady-state level of fission-associated proteins such as dynamin-related protein 1 (DRP1) and mitochondrial fission 1 protein (Fis1) (Figure 6B).

### 2.6. DEA Impeded Mitochondrial Energy Production in the BC Cell Lines

To determine the effect of DEA treatment on the energy metabolism of the BC cell lines, we used a live cell respirometer to measure the cellular oxygen consumption rate (OCR) and extracellular acidification rate (ECAR), which are indicators of oxidative and fermentative ATP production, respectively. We treated the MCF-7 and 4T1 cells with 0, 5, or 10 µM DEA for 6 h before the bioenergetics assay. Parallel to OCR, ECAR was also monitored, and the instrument calculated multiple parameters of cellular energy metabolism from the original recordings (Figure 7). At the concentration of 10 µM, DEA significantly decreased basal respiration, maximal respiration, non-mitochondrial oxygen consumption, mitochondrial ATP production, coupling efficiency, and spare respiratory capacity in both BC cell lines (Figure 6). Five µM DEA increased the proton leak in both cell lines (Figure 7). On the other hand, DEA did not affect ECAR (Figure 7), indicating that the drug did not interfere with the glycolytic machinery in either BC cell line.

### 2.7. COX-2 Inhibition Potentiated DEA’s Anti-Neoplastic Effect in the TN BC Cell Line

Elevated expression of COX-2—associated with progressive tumor growth and resistance [8]—may have accounted for the increased resistance to DEA treatment of 4T1 TNBC cells compared to the HR+ MCF-7 cells. To test this possibility, we treated both cell lines with 0–15 µM of DEA for 24 h in the presence and absence of 20 µM of the COX-2 inhibitor celecoxib before measuring the viability of the cells using the sulforhodamine B (SRB) assay. The SRB assay is based on protein content rather than metabolic activity. Therefore, it is recommended for determining the cytotoxicity of substances that can have mitochondrial effects [44]. DEA reduced the viability of both BC lines in a time and concentration-dependent way (Figure 8A). However, as expected, the viability loss caused by DEA treatment was higher for the MCF-7 than for the 4T1 cell line (Figure 7A), indicating a higher treatment sensitivity for the former cell line. However, when COX-2 was inhibited by celecoxib, DEA’s effect on the viability of 4T1 cells was significantly more pronounced (Figure 8B) in contrast to MCF-7 cells (Appendix A), suggesting that COX-2 activation may have contributed to the resistance of 4T1 cells to DEA treatment.

Colony formation assay utilizes lower drug concentrations and longer exposure times; therefore, it represents a situation more similar to the therapeutic one than the viability studies. Accordingly, we tested the effect of COX-2 inhibition on DEA’s anti-neoplastic effect using this method, too. We treated the MCF-7 and 4T1 cells with 0 to 2 µM DEA in the presence or absence of 5 µM celecoxib for 7 days before quantifying colony formation. Even at the concentration of 1 µM, DEA significantly suppressed colony formation in both cell lines (Figure 9). The TNBC line 4T1 demonstrated higher resistance against the treatment, since 2 µM DEA almost eradicated MCF-7 colony formation while it induced about a 50% decrease only in the formation of 4T1 colonies (Figure 9). However, as in the case of the viability study (Figure 8), 5 µM celecoxib augmented the effect of DEA on the 4T1 cells, and the combined treatment decreased colony formation in this cell line close to the level of the one observed in the MCF-7 line (Figure 9). In contrast, celecoxib did not affect DEA’s effect on MCF-7 colony formation (Figure 9).

## 3. Discussion

Targeting tumor metabolism seems an evident possibility, since aggressive cancers have a delicately balanced metabolism, which answers the seemingly impossible challenge of rapid proliferation in an environment of low oxygen and nutrient availability [45]. Mitochondria actively participate in all stages of cancer development from carcinogenesis via tumor survival and therapy resistance to metastasis formation [46,47], and most cancer types of high clinical grade rely on mitochondrial ATP synthesis for energy production [48,49]. Accordingly, drugs that significantly interfere with mitochondrial energy production may have therapeutic value in these tumors [50]. Based on its mitochondrial effects in B16F10 melanoma cells [34], DEA fulfills this criterion. Furthermore, and in agreement with our previous results on T24 bladder and HeLa cervix carcinomas and B16F10 melanoma [31,32,33], DEA at low µM concentrations reduced the viability of the BC cell lines MCF-7 and 4T1 in a concentration- and time-dependent manner (Figure 8A). However, in vitro cell culture experiments translate very poorly to human studies; therefore, the therapeutic dose of DEA should be determined in future animal experiments. Accordingly, the mouse 4T1 rather than the human MDA-MB-231 TNBC cell line was used in the present study.

Mitochondria regulate cellular survival via ATP production, reactive oxygen species (ROS) generation, and the intrinsic apoptotic pathway [51]. These pathways mutually regulate each other. The oxidative phosphorylation produces most of the cellular ATP, and it is one of the major source of ROS [52]. On the other hand, apoptosis depends on ATP as an energy source, while the energy shortage results in ΔΨ_m_ loss, which initiates apoptosis via the release of pro-apoptotic intermembrane proteins, such as cytochrome c, apoptosis-inducing factor, and endonuclease G [53]. DEA induced predominantly apoptotic cell death in both BC cell lines (Figure 1 and Figure 4), which was demonstrated by fluorescent staining of the phosphatidylserine residues in the outer face of the cell membrane (Figure 1), decrease in the Bcl-2/Bax ratio, activation of caspase 3, and cleavage of PARP-1 (Figure 4); all of them are hallmarks of apoptosis [54]. In agreement with its TN phenotype [36], the 4T1 cell line demonstrated higher resistance against DEA treatment than the HR+ MCF-7 cell line in these experiments (Figure 1, Figure 2, Figure 3 and Figure 4 and Figure 8A).

The proliferation rate and formation of brain, liver, and lungs metastases are much higher in the TN than in other types of BC [55]. In full agreement with these data, we found that 4T1 cells were more resistant toward DEA treatment than the MCF-7 cells in colony formation (Figure 9) and invasive growth (Figure 2 and Figure 3) experiments, too. Additionally, the findings showed that DEA decreased colony formation below 50% of the control at a concentration of 2 µM during a seven-day exposure (Figure 9), and at 10 µM, it abolished invasive growth (Figure 2) during a 12 h treatment, indicating that DEA’s therapeutic concentration (to be determined in vivo) may not exceed the DEA concentrations, which were observed during human AM therapy [24].

Constant proliferation under chronic ischemic conditions put an extra metabolic burden on cancer cells. They have to fine-tune their metabolism to meet this challenge [52,56], which makes them vulnerable against drugs that interfere with their metabolism [52]. DEA may represent such a drug candidate, since it decreased the ΔΨ_m_ in a concentration-dependent manner in both BC cell lines (Figure 5) similarly as it did in isolated liver and rat mitochondria [35] and in the B16F10 melanoma line [34].

One of the essential roles of the ΔΨ_m_ in cellular survival is the regulation of mitochondrial network dynamics that have a role in mitochondrial biogenesis and quality control, retrograde signaling, and meeting cellular energy and metabolic demands [57,58]. Mitochondrial fusion and fission processes are mediated by large GTPases; Mfn 1 and 2 and OPA1 for the fusion and Drp1 for the fission [21]. The latter is regulated by phosphorylation and recruited to the mitochondria by Fis1 [21]. Balance of the fusion and fission processes is maintained by intracellular signaling [58], but the fusion is prevented when the ΔΨ_m_ is too low [57,58]. Therefore, in many cancer types such as in astrocytomas, prostate cancers, and breast, colon, and hepatocellular carcinomas, mitochondrial fragmentation is a common feature [59,60]. Fragmented mitochondria are more prone to damage and readily eliminated by mitophagy, leading to a reduced mitochondrial copy number [21]. DEA induced fragmentation of the mitochondria in both BC cell lines (Figure 6) that could contribute to its anti-neoplastic properties. Since DEA depolarized the ΔΨ_m_ (Figure 5), it seems logical that the DEA-induced mitochondrial fragmentation was caused by inhibiting fusion that was supported by the release of OPA1 in the case of B16F10 melanoma previously [34]. In contrast, in the BC cell lines, mitochondrial fragmentation seemed to be caused by promoting fission, as it was supported by the increased Drp1 expression that was accompanied by its decreased inhibitory phosphorylation (Figure 6). It seems likely that DEA interacted with a yet to be identified regulatory element located to the mitochondria and induced mitochondrial fragmentation by shifting the balance of fusion and fission. DEA’s target has to be localized to the mitochondria, since DEA induced mitochondrial fission in isolated, Percoll gradient-purified mitochondria [35].

Prolonged activation of intracellular pro-survival signaling cascades, such as the phosphatidylinositol 3-kinase–Akt pathway, has been shown to significantly enhance cancer progression. Akt promotes cell survival and proliferation by suppressing apoptosis and stimulating cell cycle advancement [61,62]. The poor prognosis of various tumors is often associated with the constitutive activation of Akt [61,63], which phosphorylates and thereby inactivates pro-apoptotic proteins such as Bad [62]. Accordingly, dysregulation of the Akt signaling pathway is one of the most frequent oncogenic aberrations of TNBC too [64]. We demonstrated that Akt activation [61,65] was reduced dose dependently by DEA treatment in both BC cell lines. Additionally, in the HR+ MCF-7 cell line, the level of phosphorylated Akt that is the baseline activation of Akt was significantly lower than in the TN 4T1 cell line (Figure 4). The decreased Akt activation together with the aforementioned compromised mitochondrial functions may account for the differential apoptosis-promoting effect of DEA among the BC cell lines investigated (Figure 1).

Inflammatory cells and inflammatory cell mediators are prominent components of the microenvironment of tumors [14]. One of the most important immunomodulatory agents found in tumors is COX-2. It is associated with indicators of poor prognosis such as lymph node metastasis, poor differentiation, and large tumor size [66,67,68], making COX-2 the most commonly studied anti-inflammatory target in cancer therapy. The selective COX-2 inhibitor celecoxib in monotherapy and in combination with aromatase inhibitors proved to be effective in metastatic breast cancer by reducing breast tumor size and area [16,69]. In recent decades, COX-2 overexpression has been implicated in therapy resistance of various human cancers, including breast cancer [70,71,72,73]. Although hardly expressed in healthy tissues, COX-2 is highly inducible and can be rapidly upregulated in response to various pro-inflammatory agents, including cytokines, mitogens, and tumor promoters [74]. In agreement with these findings, COX-2 level was hardly detectable in the MCF-7 cells, and it was not affected by DEA treatment, while the metastatic 4T1 cell line expressed COX-2, which was elevated by DEA treatment in a concentration-dependent manner (Figure 4). Since the upregulation of COX-2 has been implicated in cancer therapy resistance [8], the latter result raised the possibility that DEA decreased its own anti-neoplastic effects in the 4T1 cell line, thereby creating the difference between the two BC cell lines in response to DEA treatment. Indeed, celecoxib potentiated DEA’s effect on viability (Figure 8B) and colony formation (Figure 9) of the 4T1 cells only. Recent studies demonstrated that during radiotherapy-induced apoptosis, caspase 3 activation led to COX-2-mediated production of prostaglandin E_2_, eventually resulting in treatment resistance [75]. DEA treatment led to caspase 3 activation in both BC cell lines (Figure 4) but upregulation of COX-2 expression in the 4T1 cell line only (Figure 4). It is likely that the DEA-induced caspase 3-assisted mitochondrial apoptosis (Figure 4) resulted in COX-2-mediated resistance in the 4T1 cell line, which expressed it. Accordingly, the difference in COX-2 expression accounted for, at least partially, the differential anti-neoplastic effects of DEA. These results also suggest that co-treatment with COX-2 inhibitors can increase the efficacy of DEA and significantly reduce therapy resistance.

## 4. Conclusions

Regardless of the mechanism, the available data indicate that DEA by interacting with a mitochondrially localized target or targets can modulate mitochondrial functions and definitely can induce the predominantly apoptotic cell death of BC cells. Although less effectively than in the HR+ BC line, DEA at low micromolar concentrations exerted effective anti-neoplastic effects in the highly treatment-resistant 4T1 TNBC cells line. Furthermore, COX-2 upregulation accounted for most of the DEA resistance by the 4T1 line that was counteracted by inhibiting COX-2’s enzymatic activity. Accordingly, considering that within the suggested safety limits, the drug does not have therapy-restricting side effects, the safety concerns might not hinder the introduction of DEA into clinical studies.

## 5. Materials and Methods

### 5.1. Materials

DEA was kindly donated by Professor Andras Varro (Department of Pharmacology and Pharmacotherapy, University of Szeged, Szeged, Hungary). All other materials if not indicated otherwise were from Sigma-Aldrich (St. Louis, MI, USA). All antibodies were from Cell Signaling Technology (Beverly, MA, USA). The following primary antibodies were used: anti-OPA1, anti-Mfn1, anti-Mfn2, anti-Drp1, anti-phospho-Drp1(Ser637), ant-Fis1, anti-Akt, anti-p-Akt, anti-Bad, anti-p-Bad, anti-Bax, anti-Bcl-2, anti-PARP, anti-caspase-3, anti-p53, anti-COX-2 (1:500 dilution), anti-actin (1:2000 dilution).

### 5.2. Cell Cultures

MCF-7 and 4T1 cell lines were from the American Type Culture Collection (Manassas, VA, USA). Both cell lines were split twice a week and were maintained as monolayer adherent cultures under standard conditions (5% CO_2_, 37 °C). MCF-7 cells were cultured in RPMI 1640 media supplemented with 10% (*v*/*v*) fetal bovine serum (FBS) and 1% penicillin–streptomycin mixture (Life Technologies, Darmstadt, Germany). 4T1 cells were cultured in RPMI 1640 media supplemented with 10% FBS, 1% penicillin–streptomycin mixture, glucose, pyruvate, and sodium bicarbonate.

### 5.3. Cell Viability Assay

MCF-7 and 4T1 cells were seeded in 96-well plates in quintuplicates at a density of 10^4^ cells, respectively. After an overnight acclimation, the cells were treated with 0 to 15 µM DEA for 24 or 48 h; then, the cells were rinsed with phosphate-buffered saline (PBS) and were fixed in 100 µL of chilled 10% trichloroacetic acid solution. Following 30 min incubation at 4 °C, the plates were rinsed five times with distilled water and then were dried overnight at room temperature. Then, 70 µL of 0.4% SRB (Sigma-Aldrich, St. Louis, MI, USA) prepared in 1% acetic acid were added to each well for 30 min at room temperature. Afterwards, the solution was discarded, and the plates were washed five times with 1% acetic acid and were dried at room temperature for 3 h. Then, 200 µL of a 10 mM tris(hydroxymethyl)aminomethane base was added to each well, and the plates were agitated at room temperature on a plate shaker for 30 min to solubilize the bound SRB. Absorbance was measured simultaneously at 560 and 600 nm with a GloMax^®^-Multi Instrument (Promega, Madison, WI, USA). Optical density (OD)_600_ was subtracted as a background from the OD_560_. The experiments were repeated five times.

### 5.4. Apoptosis Assay

To detect live, early apoptotic, late apoptotic,a and dead cells, a MUSE Annexin V & Dead Cell Kit (Luminex Corporation, Austin, TX, USA) was used. The experiments were carried out according to the manufacturer’s protocol. MCF-7 and 4T1 cells were plated at a starting density of 10^5^ cells/well into 6-well plates and were treated for 24 h with 0, 5, or 10 µM DEA. After the treatment, the cells were harvested and were diluted in their medium. Then, 100 µL Annexin V reagent was added to the samples (100 µL), which was followed by 20 min incubation in a dark room at room temperature. Five thousand single-cell events were measured per sample using a MUSE Cell Analyzer device.

### 5.5. Bioenergetics Assay

OCR and ECAR were monitored by an Agilent Seahorse XFp Analyzer (Agilent, Santa Clara, CA, USA). MCF-7 and 4T1 cells were seeded into XFp cell culture 8-well miniplates at a starting density of 3 × 10^4^ cells/well. After overnight incubation, the cells were treated with 0, 5, or 10 µM DEA for 6 h. After the treatment, the medium was replaced to Seahorse XF assay Medium (Agilent, Santa Clara, CA, USA) pH 7.4 supplemented with 10 mM glucose, 1 mM pyruvate, and 2 mM glutamine. For the measurement, we used the following inhibitors: 1 µM of oligomycin, 1 µM of FCCP, and 1 µM of rotenone/antimycin A. After monitoring 15 min of basal respiration, the FoF1 ATPase inhibitor oligomycin was added to the system to assess ATP production. After another 20 min of recording, carbonyl cyanide 4-(trifluoromethoxy) phenylhydrazone (FCCP) was added that uncouples respiration and ATP synthesis, allowing the assessment of maximal respiration. After 20 min of further recording, the Complex I and Complex III inhibitor rotenone and antimycin A were administered to inhibit mitochondrial respiration completely for calculating proton leak and non-mitochondrial oxygen consumption. Non-cellular oxygen consumption was assessed in 2 wells running without cells and was subtracted from the corresponding OCR value. The OCR and ECAR data were normalized to the mg protein content determined by using a DC Protein Assay kit (Bio-Rad, Hercules, CA, USA). No other data correction was applied.

### 5.6. Immunoblot Analysis

MCF-7 and 4T1 cells were seeded in 10 cm plates at a starting density of 10^6^ cells/plate, were cultured overnight, and then were treated with 0, 5, or 10 µM DEA for 24 h. The cells were harvested in a chilled lysis buffer containing 0.5 mM sodium–metavanadate, 1 mM ethylenediamine–tetraacetic acid, and protease and phosphatase inhibitor cocktails (1:200). After boiling, the cell lysates were subjected to 10% sodium dodecyl sulfate polyacrylamide gel electrophoresis; then, the proteins were transferred to nitrocellulose membranes. After blocking the membranes in 5% bovine serum albumin (BSA) for 1.5 h at room temperature, they were exposed to primary antibodies diluted in blocking solution at 4 °C overnight. Appropriate horseradish peroxidase-conjugated secondary antibodies were used at a dilution of 1:5000 (anti-mouse and anti-rabbit IgGs; Sigma-Aldrich, St. Louis, MI, USA). Chemiluminescence generated by applying the WesternBright ECL HRP substrate (Advansta, San Jose, CA, USA) was measured using an Azure 300 (Azure Biosystems, Dublin, CA, USA) high-resolution imaging system. Pixel volumes of the bands were determined using ImageJ software. For membrane stripping and re-probing, the membranes were washed in a stripping buffer containing 0.1 M glycine and 5 M MgCl_2_ (pH 2.8) for 30 min at room temperature. After washing and blocking, the membranes were re-probed.

### 5.7. Colony Formation Assay

MCF-7 and 4T1 cells were plated at a starting density of 500 cells/well into 6-well plates. After culturing overnight, the cells were treated with 0–2 µM DEA for 7 days. Then, the cells were washed with PBS (Biowest, Nuaille, France) and stained with 0.1% Coomassie Brilliant blue R 250 (Merck KGaA, Darmstadt, Germany) in 30% methanol and 10% acetic acid. Using ImageJ software, scanned colonies were quantified.

### 5.8. Migration Assay

To assess cell motility, we used the wound-healing assay. MCF-7 and 4T1 cells were seeded into flat-bottom 6-well plates, and they were cultured to form a sub-confluent monolayer. Then, a wound was inflicted into the cell layer by using a sterile 200 μL pipette tip, and the cells were treated with 0, 5, or 10 μM DEA for up to 12 h. The wounds were imaged at 0, 6, and 12 h by an EVOS microscope (Thermo Scientific Hungary, Budapest, Hungary) at 4× magnification. The distance differences were measured using ImageJ software. The experiment was repeated twice in duplicates.

### 5.9. Measurement of Invasive Growth

MCF7 and 4T1 cells were seeded at a starting density of 9 × 10^3^/well and 5 × 10^3^/well, respectively, in an electronic microtiter plate (E-Plate^®^) (ACEA Biosciences, San Diego, CA, USA). The cells were cultured for 24 h before they were treated with 0, 5, or 10 µM DEA for 24 h, during which the impedance was measured every 5 min. The xCELLigence Real-Time Cell Analysis (RTCA) device (ACEA Biosciences, San Diego, CA, USA) was used according to the manufacturer’s protocol. The instrument was placed in a humidified incubator at 37 °C and 5% CO_2_. These experiments were repeated twice running in three parallels.

### 5.10. ∆Ψ_m_ Assay

We used the membrane potential-dependent fluorescent dye, JC-1 (Sigma-Aldrich) at the final concentration of 1 μM for determining the ∆Ψ_m_. MCF-7 and 4T1 cells were seeded on glass coverslips and cultured at least overnight before experiments. After subjecting the cells to the appropriate treatment (as indicated in the figure legend), coverslips were rinsed twice in PBS, and cells were incubated in phosphate-buffer containing JC-1 (5 mg/mL) stain for 30 min in a CO_2_ incubator at 37 °C. Images were taken with a Nikon microscope (Inverted Microscope Eclipse Ti-U Instruction, Auro-Science Ltd., Budapest, Hungary) equipped with a SPOT RT3 2Mp Monochrome camera including SPOT Advanced software, using a 20× objective. The same microscopic field was first imaged using the red channel followed by the green channels, and the resulting images were merged by Adobe Photoshop 7.0. In control experiments, we did not observe considerable bleed-through between the red and green channels. The same calibration parameters were applied for the batch of images obtained from the same experiment. For quantification, the ImageJ software was used after converting the images to grayscale. All experiments were performed in triplicate.

### 5.11. Analysis of Mitochondrial Network Dynamics

MCF-7 and 4T1 cells were seeded in ultrathin-bottomed 96-well plates and were cultured overnight. The cells were treated as indicated in the figure legends, were rinsed twice in PBS, and were incubated in PBS containing 20 nM of MitoTracker Red for 30 min in a CO_2_ incubator at 37 °C. Fluorescence images were taken via a 60× Plan Apo Lambda objective of an ImageXpress Micro 4 High-Content Imaging System (Bioscience Ltd., Budapest, Hungary). Image analysis for mitochondrial fragmentation was performed by MetaXpress High-Content Image Acquisition and Analysis Software as described [41]. Mitochondria shorter than 2 μm were considered as fragmented, while those longer than 5 μm were considered as filamentous. All experiments were performed in triplicates.

### 5.12. Statistical Analysis

Results are shown as means ± standard deviation (SD). ANOVA using the post hoc Dunnett test (single way or two-ways) was employed to calculate the concentration-dependent effects of DEA in each experiment. Statistical analyses were performed using IBM SPSS Statistics v20.0. Differences were regarded as significant at *p* < 0.05.

## Figures and Tables

**Figure 1 ijms-23-01544-f001:**
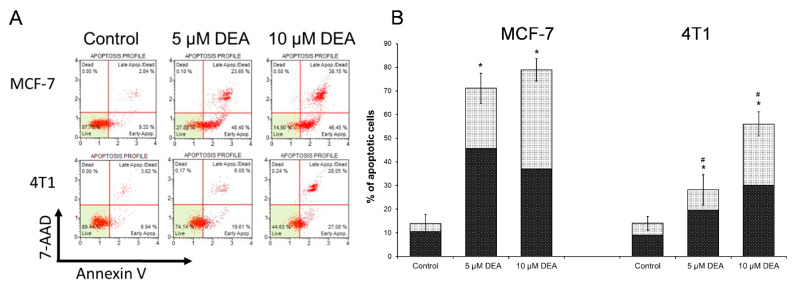
Effect of DEA on the apoptosis in BC cell lines. We treated MCF-7 and 4T1 cells with 5 or 10 µM DEA for 24 h before determining the type of cell death by flow cytometry using the Muse™ Annexin V & Dead Cell Assay kit. The results are presented as representative dot plots (**A**) and bar diagrams (**B**) of three independent experiments. The bars (**B**) indicate the sum of early (dark bars) and late (light bars) apoptosis expressed as percentage of the total cell number, mean ± SD of three independent experiments running in at least quadruplicate parallels. Controls were treated with vehicle (0.2% DMSO). * significant difference from the control (*p* < 0.05). # significant difference from the MCF-7 parallel (*p* < 0.05).

**Figure 2 ijms-23-01544-f002:**
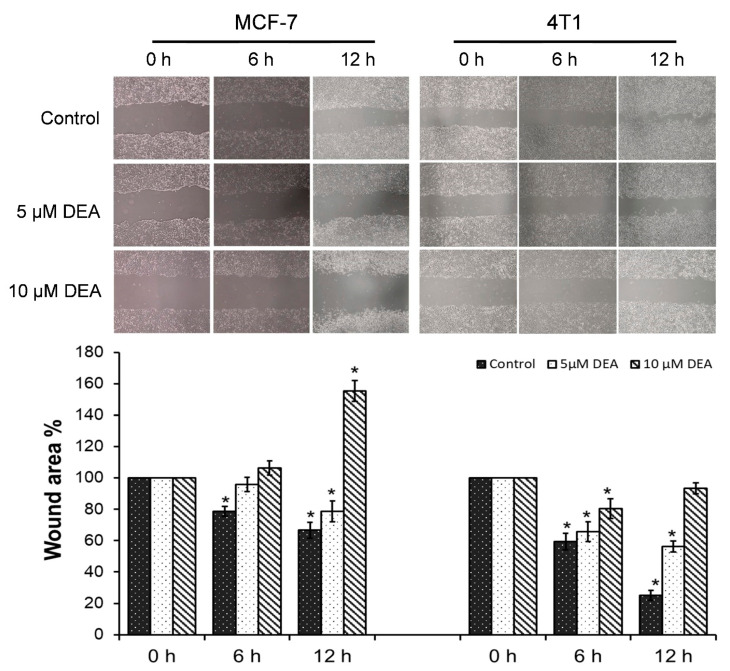
Effect of DEA on wound healing in BC cell lines. We inflicted a wound into semi-confluent cultures of MCF-7 and 4T1 cells, and treated them with 0, 5, or 10 µM DEA for 12 h. The data are presented as representative images of the wounds taken at 0, 6, and 12 h and the wound area is the percentage of untreated plates at the 0 h time-point; results are the mean ± SD of two independent experiments running in duplicates. * significant difference from the untreated control (*p* < 0.05).

**Figure 3 ijms-23-01544-f003:**
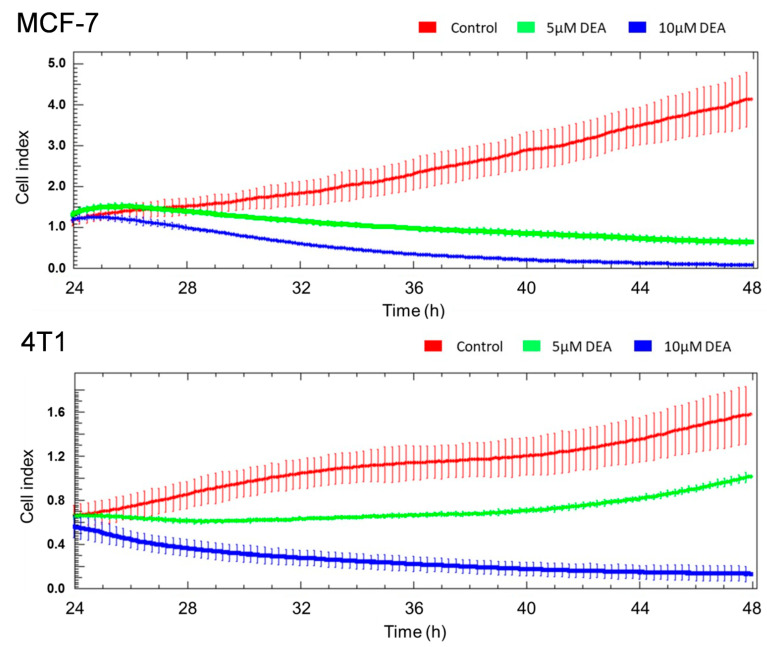
Effect of DEA on invasive growth of BC lines. MCF-7 and 4T1 cells were cultured in the presence of 0 (red line), 5 µM (green line), or 10 µM (blue line) DEA for 24 h, while the cell index was monitored by an RTCA system every 5 min. The results are presented as original recordings, mean ± SD of two independent experiments running in triplicates.

**Figure 4 ijms-23-01544-f004:**
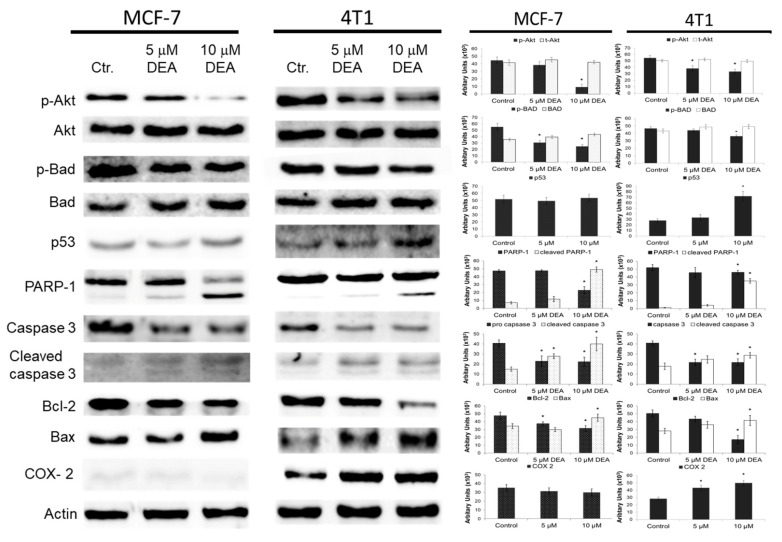
Effect of DEA on steady-state level of regulator and marker proteins of the cell death process in the BC cell lines. We evaluated the steady-state levels of p-Akt, Akt, p-Bad, Bad, Bcl-2, Bax, p53, PARP-1, caspase 3, cleaved caspase 3, and COX-2 in the MCF-7 and 4T1 cells treated with 0, 5, or 10 µM DEA by immunoblot analysis. Results are presented as representative immunoblots and pixel densities of the bands, mean ± SD of three independent experiments. * Significant difference from the control (*p* < 0.05). Additional representative immunoblots for p-Akt, Akt, p-Bad, Bad, Bcl-2, Bax, PARP-1, and COX-2 are presented in Appendix A.

**Figure 5 ijms-23-01544-f005:**
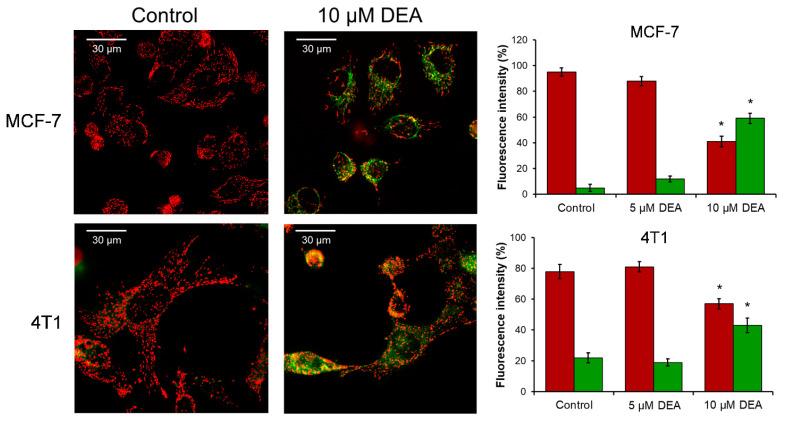
Effect of DEA on ΔΨm in BC cell lines. MCF-7 and 4T1 BC lines were treated with 0, 5, or 10 µM DEA for 3 h before loading them with JC-1 dye and taking microscopy images. The data are presented as representative merged images of the control and 10 µM DEA treated cells in the red and green channels, and as a percentage of the total fluorescence intensity in the bar diagram. Results are mean ± SD of three independent experiments. * significant difference from the untreated control (*p* < 0.05).

**Figure 6 ijms-23-01544-f006:**
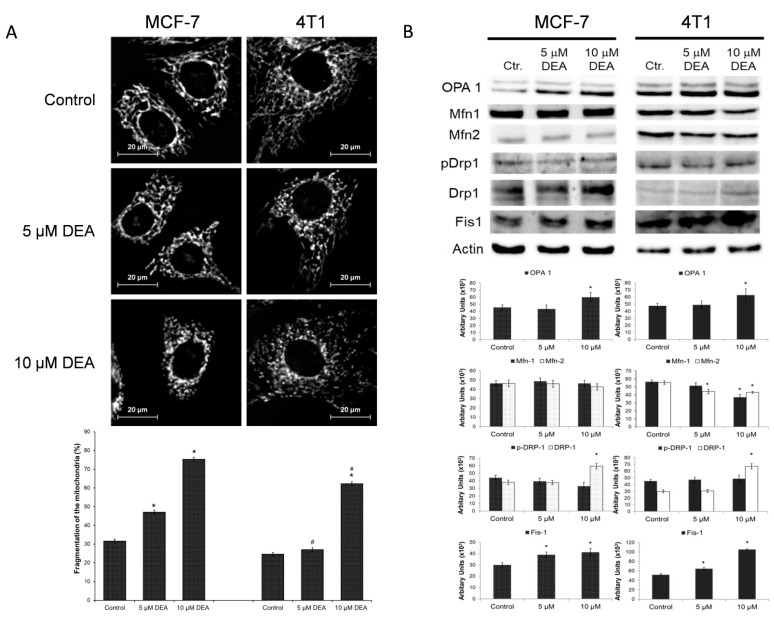
Effect of DEA on mitochondrial network dynamics in BC cell lines. MCF-7 and 4T1 cells were treated with 0, 5, or 10 µM DEA for 3 h before loading them with MitoTracker Red dye and taking microscopy images. Mitochondrial fragmentation was determined as described in [41]. The data are presented as representative images and as percentage of fragmented mitochondria (**A**), mean ± SD of three independent experiments. Additional representative images are presented in Appendix A. Separately, homogenates of identically treated cells were subjected to immunoblot analysis (**B**). The data are presented as representative immunoblots and as pixel densities of the bands, mean ± SD of three independent experiments. * significant difference from the untreated control (*p* < 0.05); # significant difference from the MCF-7cells (*p* < 0.05).

**Figure 7 ijms-23-01544-f007:**
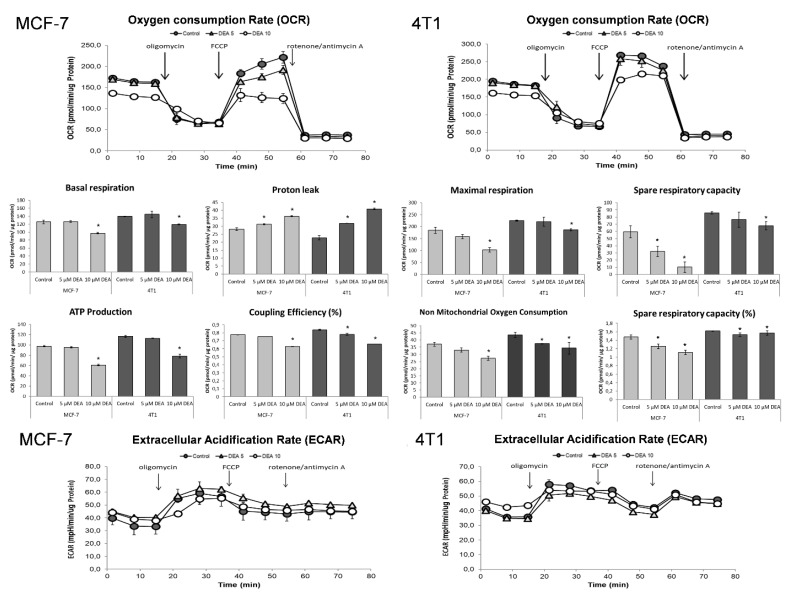
Effect of DEA on energy metabolism of the BC cell lines. MCF-7 (light bars) and 4T1 (dark bars) BC cells were treated with 0 (filled circles), 5 (triangles), or 10 µM DEA (open circles) for 6 h before monitoring OCR and ECAR for 75 min. The FoF_1_ ATP synthase inhibitor oligomycin, the uncoupler FCCP, and the respiratory inhibitors rotenone and antimycin A were added at 15, 35, and 55 min of the respiratory measurement. OCR recordings. Data are presented as representative original recordings, and as parameters, means ± SD of three independent experiments running in two replicates. OCR and ECAR data were normalized to mg protein content. * significant difference from the untreated control (*p* < 0.05).

**Figure 8 ijms-23-01544-f008:**
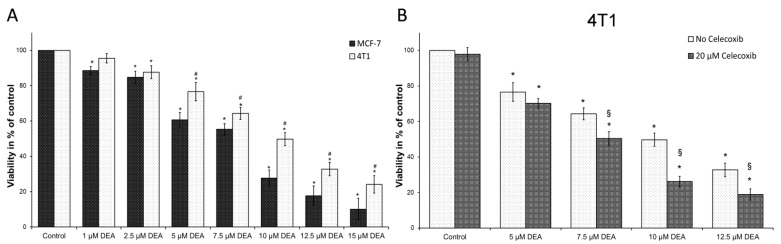
Effect of DEA on the viability of BC cell lines. (**A**): We treated MCF-7 (dark bars) and 4T1 (light bars) cells with 0 to 15 µM DEA for 24 h. (**B**): We treated 4T1 cells with 0 to 12.5 µM DEA in the absence (light bars) or presence (dark bars) of 20 µM celecoxib for 24 h. Viabilities were assessed using the SRB assay and were presented as percent of the untreated control, means ± SD of three independent experiments performed in at least quadruplicates. * significant difference from the untreated control (*p* < 0.05); # significant difference from the MCF-7 cells parallel (*p* < 0.05); § significant difference from the no celecoxib parallel (*p* < 0.05).

**Figure 9 ijms-23-01544-f009:**
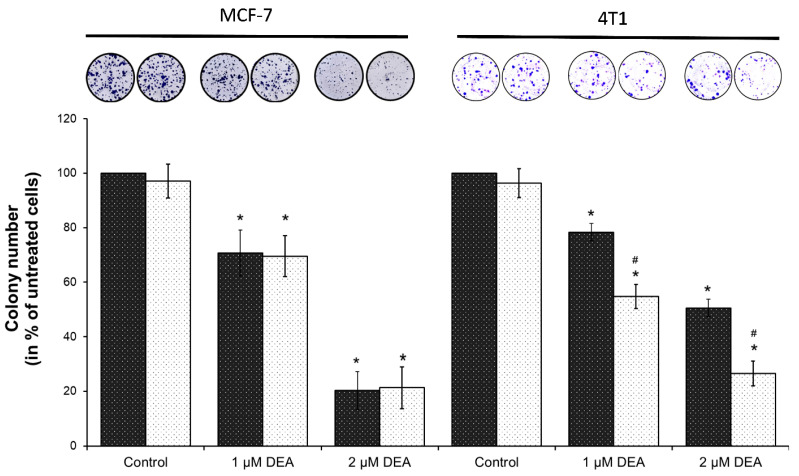
Effect of DEA and celecoxib on colony formation of BC lines. We treated MCF-7 and 4T1 cells with 0 to 2 µM DEA in the absence (dark bars) or presence (light bars) of 5 µM celecoxib for 7 days before determining colony formation. The data are presented as representative images of stained colonies in culturing plates and colony numbers are presented as the percentage of untreated plates, mean ± SD of three independent experiments. * significant difference from the untreated control (*p* < 0.05). # significant difference from the no celecoxib parallel (*p* < 0.05).

## Data Availability

Data is contained within the article or Appendix A.

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
