# Peer review of "Involvement of Mitochondrial Mechanisms and Cyclooxygenase-2 Activation in the Effect of Desethylamiodarone on 4T1 Triple-Negative Breast Cancer Line"

_ijms, 2022, doi:10.3390/ijms23031544_

Round 1

Reviewer 1 Report

Overall, the presented manuscript presents several parameters of DEA effect on mitochondrial metabolism but lacks deeper importance or applicability. There is no logical connection to COX2 line of evidence, which is very weak itself.

Introduction - connection to COX2? explain if COX2 is likeky expected to be overexpressed after drug treatment? is that a general rule in cancer?

The paragraph in the intro presents rationale about DEA/AM toxicity but how is it relevant to cancer therapy and aim of the paper?

Apoptosis induction: late vs early apoptosis – the legend should be included in the figure 1B. In the figure legend please structure into 1a vs 1b so the description of the graph his found right away.

Figure 3 – apoptosis markers. I would not consider decrease in phosphorylation of BAD as significant, please provide additional western blots repeating the experiment to show reproducibility of observation. For that matter please provide additional western blots of the key observations that you dexcribe (PARP-1, Bax, phospho-AKT). The COX-2 levels are interpreted in a way that it is important difference between TNBC and ER+ cell lines, although there are only two cell lines used. Please repeat the western blot with two cell lines in one blot side by side so we can see the real difference and consider to use other cell lines to draw clear conclusion regarding expression levels in TNBC and ER+ cell lines in response to DEA treatment.

JC-1 measurement; so the cells were stained for 30 minutes and then treated with DEA in standard medium? Is there any time-wise comparison of the JC-1 signal prior treatment, or are the presented pictures parallel samples? So authors suggest there is a depolarization of deltaPSIm due to DEA treatment, but control samples appear to be hyperpolarized. Usually, JC-1 staining of any control cells looks green with red spots, similarly to 10uM DEA treatment. I am afraid that control cells are overly stained which would distort the results.

Mitotracker red measurement; the quality of figures is not adequate to judge if there is any mitochondrial fragmentation. In JC-1 figures the fragmentation is not apparent. Please provide additional high-quality pictures that show fragmentation of mitochondrial network. Similarly, there is a Drp1 increase but with deltaPSIm changes OPA1 is usually fragmented into shorter isoforms, I am surprised there are not big changes in OPA1 levels although authors detected them by densitometry.

Explanation of methods – bioenergetics measurement – is too wordy and should not be in the Results section, but in MM section. I presume that the readership  is familiar with oligo-FCCP-rot/AA protocol. I suppose that the results support depolarization of mitochondrial membrane potential measured by JC1.

COX2 section. I don’t see the point of COX2 inhibition in combination with DEA inhibitor, I think it is obvious that the cells expressing COX2 would be more sensitive to celecoxib than cells not expressing. What is the point of the experiments and importance of the results? How does this experiment relates to mitochondrial metabolism except for SRB assay which is very unspecific?

Discussion  is way too long and does not discuss main results in the broader context, it simply repeats results from the section. Authors should think about better way to construct discussion. There are unnecessary parts that do not actually discuss any results.  

Author Response

We would like to thank the reviewer his/her thorough evaluation of the manuscript. We do believe that his/her contribution has increased the scientific value of the study considerably.

  1. We regret that the reviewer does not regard this study to be important or applicable. We think that the antineoplastic properties of DEA, which were supported by previous in vitro and in vivo experimental evidences (Refs 31-34) together with the profound knowledge about its clinical safety features makes DEA a quite promising therapeutic tool. Furthermore, one of the reviewers of Ref 34 liked the idea of repurposing a stable and abundant metabolite of a drug in clinical use for cancer therapy too in an extent that he/she thought it would have been worth publishing the concept as a hypothesis even without any experimental support. Nevertheless, we tried to clarify the importance and applicability of DEA’s various effects throughout the manuscript.

A number of preclinical reports indicated the role and significance of COX-2 overexpression and increased activity in progressive tumour growth and therapy resistance. Accordingly, we felt studying the interaction between celecoxib and DEA to be justified. As it turned out, the TN 4T1 line was more resistant to DEA treatment than the HR+ MCF7 line. Furthermore, DEA treatment massively induced the low COX-2 expression in 4T1 line, while the MCF7 line, which does not express COX-2, was not affected. Additionally, inhibiting enzymatic activity of the DEA-induced COX-2 by celecoxib sensitised the 4T1 cells to the DEA treatment while celecoxib alone did not affect the viability of the cells (Fig 8B) indicating that elevated expression of COX-2, which was observed in the 4T1 cells only (Fig 4) likely contributed to the aforementioned resistance. 

  1. COX-2 likely via prostaglandin production promotes progressive tumour growth and therapy resistance (Refs 8, 11-16). Tumours of more aggressive phenotype express COX-2 at a high extent without any drug treatment even when their less aggressive equivalents do not express this enzyme or express them at a much lower level. However, chemotherapy can induce COX-2 expression, which is implicated in the development of therapy resistance (Ref 8, 15).
  2. Antineoplastic drugs with the exception of those of targeted therapy are generally toxic to normal cells, but accumulate in higher extent in cancer cells or affect them more severely. Accordingly, tissue accumulation and cytotoxicity, although not necessarily, could indicate antineoplastic potential in a drug that gave us the initial idea of investigating whether DEA has such potential (Ref 31). Nevertheless, we deleted the paragraph about the side-effects of amiodarone, which are clearly not relevant in tumour chemotherapy.
  3. Ad Fig 1: We have indicated which part of the legend refers to Fig 1A and B as suggested by the reviewer.
  4. Ad Fig 4: In an attached ZIP file (RequestedBlots), we provided all original immunoblots requested by the reviewer. We repeated the requested experiments and presented the parallels side by side on the same gel and presented the data as Supplementary Figure 1.

However, we found the time frame for revision too narrow for thawing and expanding additional cell lines, and preforming the experiments on them too. Rather, we have opted to restrict our conclusions to 4T1 and MCF7 lines only.  

  1. Ad Fig 5: For assessing DEA’s effect on the ΔΨ, we used the same experimental protocol for each sample. Therefore, the samples can be considered as parallels. As we described in the Material and Methods section, the cells were treated with DEA in normal culturing media for the indicated time then were loaded with JC-1 before they were subjected to imaging. Formation of J aggregates during JC-1 staining depends on the dye concentration and the ΔΨ. Accordingly, it is the dynamics rather than the actual intensity ratio of red and green fluorescence that tells us how ΔΨ is changing. When intensity ratio of red and green fluorescence is balanced initially and the ratio increases, the ΔΨ is hyperpolarised and vice versa. We agree with the review that our cells were overstained, but disagree with him/her that it distorts the results. It is simply a protocol, which focuses on assessing differences in the degree of depolarisation by maximising the range of red and green fluorescence intensity ratio.
  2. Ad Fig 6: We fully agree with the reviewer that resolution of the Mitotracker Red images was low. We provided higher resolution version of the original images (Figure 6A) and presented another set of representative images of the same experiment as a Supplementary Figure 2. Additionally, mitochondrial fragmentation is quite relevant in red channel of the JC-1 images (Figure 5).

We also agree with the reviewer that ΔΨ depolarisation usually is accompanied by OMA1 mediated OPA1 cleavage, and we admit that the representative blot presented in the original Figure 5 was not a fortunate choice to demonstrate it. Accordingly, we replaced the representative OPA1 blots in the original Figure 5 (Figure 6) that reflects the densitometry results more faithfully, and presented additional blots in the RequestedBlots ZIP file as requested by the reviewer.

  1. We have moved description of the bioenergetics measurement to the Materials and Methods section as suggested by the reviewer.
  2. We would agree with the reviewer if 4T1 cells responded to celecoxib. The essence of this experiment is that celecoxib alone does not affect viability of the 4T1 cells, but sensitise them to DEA treatment (Fig 8B). COX-2 activity-mediated resistance explains, at least partially, increased resistance to DEA treatment by 4T1 vs. MCF7 cells.

The SRB assay determines protein content, which is proportional to cell number. It is suggested for assessing viability when any drug used in the experiment interferes with mitochondrial metabolic activity, which DEA clearly does.

  1. We agree with the reviewer that the purpose of the Discussion is to put the results in broader context and compare them with those of other groups. Unfortunately, we are the only group publishing about DEA’s antineoplastic effects, which hopefully will amend once we can convince clinicians to perform a proof of concept pilot human study. Accordingly, at present, we can compare the results with our previous findings only, and try to put them in a broader sense by introducing related or supportive material from the literature. In the revised version, we tried to eliminate parts that do not discuss results, and avoid repeating the results as suggested by the reviewer.

Reviewer 2 Report

The publication "Involvement of Mitochondrial Mechanisms and Cyclooxygenase-2 Activation in the Effect of Desethylamiodarone on 4T1 Triple-Negative Breast Cancer Line" is an interesting scientific study with a future clinical aspect.
All chapters of the work are properly prepared. The Materials and mMthods are described in detail. The Authors present the results in a graphic form. All figures  and photos in the publication are clear and therefore easy to analyze.
The references chapter covers the most of the references items published in the last 15 years. I think that giving historical references, i.e. from the previous century in experimental studies, is unnecessary, possibly replacing them with newer ones.
Due to the interesting topic and good quality of the manuscript, I propose to accept it for printing in IJMS in its current form.

Author Response

We would like to thank the reviewer his/her thorough evaluation of the manuscript.

We made an effort to improve the manuscript’s English, however, did not have the time to send the manuscript to a scientific proofreading company. However, we are happy to do so if the editor or the reviewer thinks it necessary.  

Reviewer 3 Report

<major comments>

  • Fig 2. authors performed experiments that anti-invasive effects of DEA on cancer cells growth with scratch test. It was classical and nice tools but was old designs to further 'valuable' data. Even, If so, authors should present statistic densitomety data of cell confluencies because cell growth density was significantly increased at high dose of DEA. So this reviewer suggest that Fig 2 exchange to other data such as 3D invasion tests.
  • authors must discuss that why DEA was effective also in MCF-7 cells? HER2 was crucial mechanism or meaningless molecules on action mechanisms of DEA? then other receptors are important targets of DEA?

<minor comments>

  • authors mentioned "B16F10" at many point  even results. Is it important?

Author Response

We would like to thank the reviewer his/her thorough evaluation of the manuscript. We do believe that his/her contribution has increased the scientific value of the study considerably.

  1. We agree with the reviewer’s concern. Therefore, we performed a 3D invasive growth test to support findings of the wound healing experiments as suggested. We presented the results in a new figure (Figure 3), and added text to the Materials and Methods, Results and Discussion sections accordingly. Although it is not evident on the representative images, DEA did not increase cell densities. Rather, 10 µM of DEA tended to kill MCF7 cells at the edges indicated by widening of the scratch (Fig 2).
  2. We do not think that DEA’s antineoplastic effect has anything to do with hormone or growth factor receptor mediated mechanisms. Additionally to the BC lines, DEA had effective cytotoxicity in bladder and cervix carcinoma and melanoma lines. Accordingly, its cytotoxicity is likely based on processes, which are common in these cell lines such as mitochondrial mechanisms. In the present study, we used the HR+ MCF7 line simply as a cancer line of less aggressive phenotype as the TN 4T1 BC line.
  3. B16F10 is an aggressive, metastasising melanoma cell line. In fact, DEA had the most prominent antineoplastic effect on B16F10 line among the cell lines we have studied up to now. Therefore, we referred to the results on B16F10 line for comparative purposes.  

Round 2

Reviewer 1 Report

I see some improvements in the text but the major objection remains.

Mitotracker figures – put zoomed figures (insets zoomed on details), although figures are better but fragmentation itself cannot be seen.

Actine in WB cations – change to Actin.

I see now the point of DEA treatment with celocoxib experiment, I must suggest that figure legend in 8B captions is changes into DEA (white) vs DEA + 20uM Celocoxib (black). Also, figure 8A is very confusing as you compare MCF7 to 4T1 cells but the graph title says MCF-7. Please add experiment of DEA + celecoxib treatment for MCF7 to demonstrate similar/dissimilar effect of the combined treatment.

However, I still lack the connection of COX-2/DEA part to the rest of the article. The rationale is not elaborated enough.  For me there are two stories there with no connection except that COX-2 expression is slightly induced by DEA treatment in 4T1 cell line along with changes in mitochondrial parameters? Either you include very strong argumentation to support your line of evidence, otherwise I do not see the point of the publication even though experiments are interesting. Maybe I am missing something important which means that the rationale is not strong enough.

Author Response

Response to Reviewer 1

We would like to thank the reviewer his/her thorough evaluation of the manuscript. We do believe that his/her contribution has increased the scientific value of the study considerably.

  1. We are glad that the reviewer acknowledged some improvement in the manuscript’s quality, but we do not understand why his/her major objection did remain the same. Comparing his/her round 1 and round 2 evaluations, we would think that that objection shifted rather than remained the same.
  2. We fully agree with the reviewer about the quality of Figure 6A. However, Figure 6 is already crowded, and providing expanded inserts would have increased its complexity much further. Nevertheless, to comply with the reviewer’s request at least in essence, we replaced the images in Figure 6A to a new set imaged with a 60x rather than a 20x objective, and omitted the colour information for better contrast and resolution.
  3. We have changed Actine to Actin in all figures.
  4. We are indeed happy that we have succeeded in clarifying point of the celecoxib — DEA co-treatment experiment in the revised manuscript. According to the reviewer’s request, we revised the legend of Figure 8, and presented the results of DEA + celecoxib treatment on viability of MCF-7 cells. However, since the results were essentially negative, we presented these results as Supplementary Figure 3 rather than a panel of Figure 8.
  5. We do not share the reviewer’s view about connection of the COX-2/DEA part to the rest of the article. (i) We cited references demonstrating that COX-2 overexpression was indicated in therapy resistance of various types of cancer including breast cancer. (ii) We showed that the breast cancer lines we used differentially expressed COX-2, and the DEA treatment elevated COX-2 expression in the triple-negative 4T1 cell line only (Figure 4). (iii) Additionally, we demonstrated that the 4T1 cell line was more resistant than the MCF-7 cell line to the DEA treatment (Figure 8A), which resistance was eliminated by the selective COX-2 inhibitor celecoxib (Figure 8B, 9). (iv) In the present revised manuscript, we cited a reference providing experimental data supporting the role of the caspase 3 — COX-2 — prostaglandin E2 pathway in the therapy resistance of cancer cells. (v) In the Discussion section, we suggested that the mitochondrial damage caused by the DEA treatment resulted in caspase 3 activation (Figure 4) that according to the aforementioned mechanism could (at least partially) lead to the COX-2-mediated resistance of 4T1 cells to the DEA treatment.

We do believe that the argument in the previous paragraph is persuasive in justifying inclusion of the COX-2 data in the present study. Additionally, we are positive that any further argument would be over-interpretation of the available data.  

Reviewer 3 Report

This paper has been improved.

Author Response

Response to Reviewer 3

We would like to thank the reviewer his/her positive opinion about the revised manuscript. We have tried to further improve the quality in the 2nd round of revision.

Round 3

Reviewer 1 Report

Thank you for another round of revisions. I honestly don´t know how to react. I still do not think that rationale is strong enough to be published in the rather well-rated journal (but the preferred content depends of course on the journal policy). Experiments are of course performed well and professionally, but the results and its significance is not vast. The added graph of MCF7 combination treatment DEA+CE does not support the rationale, nor mitochondrial line of evidence. 

From the practical perspective, the experiments and presentation is OK, the scientific significance and the interpretation of results is less strong.